# The Situation and Influencing Factors of Depression and Anxiety in Patients of Hemodialysis during the COVID-19 Pandemic in China

**DOI:** 10.3390/healthcare11070941

**Published:** 2023-03-24

**Authors:** Dan Jiang, Xi Yu, Tian Zhong, Ying Xiao, Ling Wang

**Affiliations:** 1Faculty of Medicine, Macau University of Science and Technology, Macau 999078, China; 2The 8th Affiliated Hospital of Sun Yat-Sen University, Shenzhen 518033, China

**Keywords:** depression, anxiety, maintenance hemodialysis, COVID-19

## Abstract

Objective: To investigate the incidence of depression and anxiety among maintenance hemodialysis (MHD) patients during the regular prevention and control stage of COVID-19 in China as well as the influencing factors. Methods: A cross-sectional study including 180 patients under the treatment of hemodialysis was conducted in the hemodialysis center of the 8th Affiliated Hospital of Sun Yat-Sen University. The questionnaire regarding the subject’s general information, Self-Rating Anxiety Scale (SAS) and Self-Rating Depression Scale (SDS) was completed by the patients, and the laboratory test results were recorded. Results: The incidences of anxiety and depression were 35.6% and 38.9%, respectively, and the average scores of SAS and SDS were (48.03 ± 5.02) and (48.12 ± 5.42), respectively, in the subjects. The results showed that age, monthly income, vascular access of dialysis, feeling of pain and itching (within a week), worried about being infected by COVID-19, whether having health insurance and the levels of hemoglobin, parathyroid hormone, and phosphorus were the impact factors of both anxiety and depression in the MHD patients (*p* < 0.05 for all). Conclusion: The proportion of depression and anxiety is relatively high in the MHD patients during the regular prevention and control stage of COVID-19.

## 1. Introduction

Since the outbreak of the epidemic in 2019, the psychological status of various groups, including the general population, isolated groups, medical staff and patients, has been affected to various degrees. Many studies have been conducted on the mental health problems in these groups. For example, several studies [1,2,3,4] have been conducted on the mental health of people during lockdowns for the purpose of breaking the spread of the epidemic. They showed that when people were confined to certain environments, their mental health was adversely affected. Similar results were found in Chinese populations during the COVID-19 pandemic, which had a major impact on participants’ mental health; 53.8% of the respondents believed that the epidemic had a moderate or severe negative psychological impact; 16.5% of the respondents had moderate to severe depressive symptoms; 28.8% of the respondents had moderate to severe anxiety symptoms; 8.1% of the respondents were under moderate to severe stress levels. Most respondents worried about their family members being infected by COVID-19 (75.2%) [2]. 

Previous studies have revealed the impact of COVID-19 on anxiety, depressive, fear, denial, insensitivity, and poor sleep quality depending on individual pre-existing psychological problems and sociodemographic factors [5,6]. Previous chronic disease might play a role in the psychological impact of COVID-19 on individuals. Patients with chronic diseases were more susceptible to contracting COVID-19. They were more likely to develop critical disease with a higher mortality than the patients without chronic diseases [6,7]. Patients with chronic diseases would worry that their timely treatment and medical care would be hindered due to the limitation of epidemic prevention and control.

Similarly, chronic kidney disease (CKD) is one of the most common chronic diseases, and the patients with CKD are often accompanied by varied degrees of anxiety, depression and other psychological problems due to the long duration of dialysis and various complications. During the epidemic, patients with CKD need to continue the regular dialysis. It is difficult for them to avoid taking public transportation and staying in public areas, so the patients often worry about being infected by the virus during the trip between home and hospital and the period of dialysis, which can exacerbate mental health problems such as anxiety and depression. Depression and anxiety are often comorbid with the end-stage renal disease (ESRD), and they are associated with poor adherence to the treatment and low quality of life in the patients under dialysis [8]. At present, a large number of studies have shown that the outbreak of COVID-19 had a negative impact on the mental health of hemodialysis patients. However, there are few reports on the anxiety and depression of hemodialysis patients during the regular prevention and control stage of COVID-19. Limited studies addressed the direct impact of COVID-19 on mental health during the stable stage of the epidemic in the patients with ESRD. Therefore, this study will make the following contributions: (1) exploring the mental health status of the patients under MHD at the regular prevention and control stage of COVID-19 and analyzing the influencing factors, and (2) providing a reference for reducing anxiety and depression in MHD patients.

## 2. Subjects and Methods

### 2.1. Participants and Procedure

This study was a cross-sectional survey. The subjects were the patients who underwent long-term hemodialysis from February 2019 to June 2022 in the 8th Affiliated Hospital, Sun Yat-Sen University. A structured questionnaire was used to assess the mental health status of the patients during the regular prevention and control stage of COVID-19. The survey was conducted from 23 to 30 June 2022. To prevent the spreading of COVID-19, the preventive and control policies were set up based on the risk level at different regions in China. The local government imposed restrictions on hospitals with dialysis patients, including a series of measures for hemodialysis centers, such as forming an expert group setting to handle the patient treatment and increasing dialysis shifts and dialysis isolation wards to meet dialysis needs. Inclusion criteria for the study subjects: (1) were MHD patients; (2) agreed to participate in the research; (3) were able to read or understand the content of the questionnaire and communicate orally or in writing. Exclusion criteria: (1) had history of mental and neurological diseases, long-term alcohol or drug abuse, long-term use of opioids or tranquillizers; (2) had cognitive impairment or severe visual impairment; (3) had complications that affect the patient answering the questionnaire; (4) had critical disease. This study was approved by the ethics committee of the 8th Affiliated Hospital, Sun Yat-Sen University. A total of 180 patients were included in the study and signed informed consent.

### 2.2. Methods

#### 2.2.1. Questionnaire Survey

The self-designed questionnaire for collecting general demographic data included information on gender, age, monthly income, whether having healthcare insurance, feeling of pain, skin itching, duration of dialysis, vascular access of dialysis, and worried about being infected by COVID-19.

Self-rating anxiety scale: The Self-Rating Anxiety Scale (SAS) created by Zung [9] was used to assess anxiety. SAS is reliable, efficient, and widely used. Cronbach’s α coefficient is 0.823. The scale contains 20 items that reflect subjective feelings of anxiety, of which 15 had positive points and 5 had negative points. Each question was divided into four levels: respondents rated how they felt about the past week on a scale of 1 (none or little) to 4 (most or all of the time). The scores for none, mild, moderate and severe anxiety were 50, 50–59, 60–69 and >69, respectively.

Self-rating depression scale: The Self-Rating Depression Scale (SDS) created by Zung [9] was used to evaluate depression. This scale also consists of 20 items; the positive and negative were half and half with a 4-point Likert scale ranging from 1 (none or few times) to 4 (most or all times).

Each participant completed the questionnaire on We Chat, which is a widely used social media APP in China, including SAS, SDS and questionnaires for general demographic information. The questionnaire was anonymous to ensure the confidentiality and reliability of the data.

#### 2.2.2. Statistical Analysis

All data were analyzed by using SPSS 22.0 (SPSS Inc., Chicago, IL, USA). Continuous variables were presented as mean ± SD. Categorical variables were expressed as number and percentages (%). Differences between groups were compared using one-way ANOVA and Chi-square test. Spearman’s correlation coefficient (r) was used to assess the association between laboratory test data and levels of anxiety and depression. Statistically significant variables were further analyzed through multivariate logistic regression. Estimates of association strength were demonstrated by odds ratios (ORs) with 95% confidence intervals (CIs). Significant levels were set as *p* < 0.05 (two-tailed).

## 3. Results

### 3.1. Demographic and Clinical Parameters

Among the 180 participants, there were 96 men (53.3%) and 84 women (46.7%), 113 patients were ≥50 years old (62.8%), 84 patients had lower than 10,000.0 RMB (46.70%) monthly income, 89 patients (49.4%) underwent dialysis over three years, 105 patients were (58.3%) worried about being infected by COVID-19, 87 patients (48.3%) felt localized pain, 130 patients (72.2%) had healthcare insurance, 127 patients (70.6%) were using arteriovenous fistula (AVF) as their vascular access of dialysis, of which there were 53 cases (29.4%) with catheters and artificial blood vessels. The average hemoglobin (Hb) was 111.49 ± 15.58 g/L, albumin (Alb) was 38.06 ± 4.13 g/L, serum ferritin (SF) was 129.94 ± 167.46 ng/mL, blood calcium (Ca) was 2.26 ± 0.21 mmol/L, parathyroid hormone (PTH) was 243.73 ± 190.62 pg/mL, blood phosphorus (P) was 1.98 ± 0.64 mmol/L, serum potassium (K^+^) was 4.82 ± 0.68 mmol/L, and β_2_ microglobulin was 25.56 ± 7.95 mg/L.

### 3.2. Analysis of Anxiety and Depression of the MHD Patients

Among the 180 MHD patients, the SAS and SDS scores were 48.03 ± 5.02 and 48.12 ± 5.42, respectively. There were 64 (35.6%) and 70 (38.9%) cases of anxiety and depression, respectively, of which the majority (63, 35.0% and 66, 36.7%) consisted of mild anxiety and depression, respectively.

### 3.3. Single Factor Analysis of Anxiety and Depression among the MHD Patients

The general information and laboratory tests of the MHD patients were divided into groups according to whether they were anxious or depressed. The differences of monthly incomes, vascular access of dialysis, duration of dialysis, feeling of pain, worried about being infected by COVID-19, skin itching, and levels of Hb, PTH, and P were significant between the anxiety and non-anxiety groups as well as depression and non-depression groups (all *p* < 0.05). Possession of medical insurance and age (<50 years) also significantly varied between depression and non-depression groups (all *p* < 0.05). For details, please refer to Figure 1, Figure 2, Figure 3 and Figure 4.

### 3.4. Correlation between SAS/SDS Scores and Laboratory Tests among MHD Patients

The Pearson correlation analysis results showed that the SAS and SDS scores of MHD patients were positively correlated with PTH and P (all *p* < 0.05), but they were negatively correlated with Hb (*p* < 0.05), as shown in Table 1.

### 3.5. Analysis of Multiple Factors Influencing the SAS and SDS Scores in the MHD Patients

The above significantly different parameters were taken as independent variables, and anxiety and/or depression were set as dependent variables for stepwise regression analysis. The assignment of logistic regression was shown in and Appendix A
Table A1.The results showed that monthly incomes, vascular access pattern, feeling of pain (within a week), age ≥ 50 years, worried about being infected by COVID-19, Hb, and PTH were influencing factors of anxiety (*p* < 0.05). In the MHD patients, for the risk of anxiety, the number of individuals with higher income (≥10,000.0 RMB/Month) was 0.284 times that of those with lower income (<10,000.0 RMB/Month) (OR = 0.284, 95% CI: 0.102–0.786) (Figure 5), and the number of individuals who felt pain was 4.226 times that of those who did not feel pain (OR = 4.226, 95% CI: 1.505–11.871). The number of subjects worried about being infected by COVID-19 was 3.543 times that of those not worried about (OR = 3.543, 95% CI: 1.205–10.420), the number of those not using AVF as their vascular access of dialysis was 2.989 times that of those using AVF as the vascular access of dialysis (OR = 2.989, 95% CI:1.118–7.989). In addition, older age (≥50 years) was 0.24 times more frequent than younger age (<50 years) (OR = 0.24, 95% CI: 0.082–0.703). The risk of anxiety was increased by 0.5% (OR = 1.005, 95% CI: 1.002–1.009) for every unit increase in blood PTH level, but it decreased by 3.9% (OR = 0.961, 95% CI: 0.926–0.998) for each unit increase in Hb in the MHD patients. For details, please refer Figure 5 and Appendix A
Table A2.

Analysis of depression among MHD patients with laboratory tests showed that age, duration of dialysis, vascular access of dialysis, feeling of pain, worried about being infected by COVID-19, Hb, PTH, and P were the factors that influenced depression (all *p* < 0.05). Among the MHD patients, for the risk of depression, the longer duration of dialysis (≥3 years) was 2.668 times the shorter duration (<3 years) (OR = 2.668, 95%CI: 1.027–7.040), older age (≥50 years) was 0.276 times more frequent than younger age (<50 years), (OR = 0.276, 95% CI: 0.097–0.784), feeling pain was 5.286 times more frequent than non-feeling (OR = 5.286, 95% CI = 1.945–14.364), worried about being infected by COVID-19 was 4.423 times more frequent than not feeling worried (OR = 4.423, 95% CI = 1.549–12.631), not using AVF as vascular access of dialysis was 3.014 times more frequent than using AVF (OR = 3.014, 95% CI: 1.132–8.020). For details, please refer to Figure 6. Regarding the blood tests, the risk of depression was increased by 0.4% (OR = 1.004, 95%CI = 1.001–1.007) for every unit increase in PTH, but it decreased by 4% for each unit increase in Hb (OR = 0.960, 95% CI = 0.926–0.996) in the MHD patients. The risk of depression with abnormal blood P was 4.68 times higher than those with normal blood P (OR = 4.680, 95% CI: 1.199–18.269) in the subjects. For details, please refer to Figure 6 and Appendix A
Table A3.

## 4. Discussion

### 4.1. Anxiety and Depression of the MHD Patients during the COVID-19

Chronic kidney disease (CKD) is a global health problem characterized by high morbidity, complications, disability and mortality. The rapid increase in the number of patients with CKD and end-stage renal disease has caused a heavy social and economic burden. The prevalence of CKD in China has exceeded 10%, meaning that at least 1 in 10 people has CKD, and CKD is predicted to become the fifth leading cause of death worldwide by 2040.

The MHD patients have many psychological problems even under general circumstances. Before the COVID-19, 20% of the MHD patients suffered from mental health problems [9]. Meanwhile, 35.6% and 38.9% of investigated MHD patients presented as having anxiety and depression, respectively, even during the period of regular preventive and control of COVID-19 in our study, which is significantly higher than previously. Our findings are similar with the report conducted in China regarding the MHD patients’ mental health during the COVID-19 pandemic [10,11]. The incidence of mental problems in the present study was obviously higher than that of the Polish and German young adults (18–35 years old) (16.2% and 30% for anxiety and depression, respectively), as well as the result in India during the locked down and quarantined period of the COVID-19 pandemic (25%) [8,12,13]. Data from other countries showed that the mortality of the patients with chronic diseases was more than six times that of those without chronic diseases after being infected by COVID-19 [14]. The higher risk might be due to the coronavirus directly affecting the brain or indirectly by inducing a massive cytokine response [15].

A meta-analysis from Ko et al. [16] showed that the incidence of anxiety and depressive in patients with CKD or ESRD during the COVID-19 pandemic was higher than the non-epidemic period (26.5%). This may be explained by the high risk being exposed to the virus infection on the public transportation and public area of health service units, which made the patients more stressful. Additionally, according to the prevention and control measures, the patients were asked to reduce unnecessary hospital visits or even cancel their appointment for routine dialysis, unless there were special circumstances, wear masks throughout the dialysis process, disinfect their hands, and avoid eating and drinking in the dialysis room. These restrictions may increase the psychological burden of the patients.

### 4.2. The Influencing Factors of Anxiety and Depression in the MHD Patients during the COVID-19

Levin’s [17] previous studies in ESRD patients showed that dialysis patients were at risk of developing anxiety and depression. The current study implied that the HD patients should be considered as a highly susceptible population with a higher risk of psychiatric problems during the period of regular prevention and control of COVID-19.

Our current study showed that the feeling of pain was significantly associated with the risk of depression and anxiety. Pain is a common symptom in CKD patients. No matter whether being treated with dialysis or other treatment, pain occurs in more than 58% of patients with CKD and about 49% with moderate to severe pain [18]. On one hand, pain might be caused by the disorder of bone metabolism of the ESRD; on the other hand, the underlying disease as well as the comorbidities of the disease leading to CKD contribute to chronic pain [18], which are the factors inducing anxiety and depression. Pain, particularly chronic pain, is known to be associated with a variety of adverse outcomes such as depression, reduced quality of life, and increased hospitalization. It was reported [19] that the feeling of pain had a significantly positive correlation with the levels of anxiety and depression in all cases.

This study demonstrated that patients who worried about being infected by COVID-19 had a high risk of anxiety and depression compared with those did not worry about it. This might be explained as patients needing to maintain life and undergo regular hemodialysis. Most of the hemodialysis patients are middle-aged or elderly people, often with chronic diseases such as hypertension and diabetes, and they might have poor immune defense capacity [20,21,22,23]. Therefore, they are more susceptible to being infected by microorganisms than healthy people. During the period of the pandemic, as well as the regular prevention and control of COVID-19, the MHD patients have a high chance of being infected by the virus due to staying in public areas.

Vascular access is essential for maintaining hemodialysis in MHD patients. Vascular access among hemodialysis patients mainly includes: autogenous arteriovenous fistula (AVF), arteriovenous graft (AVG) and central vein catheter (CVC). Many studies have confirmed [24,25,26] that AVF is the optimal choice as vascular access among hemodialysis patients. If AVF could not be established, the second choice should be AVG, and then CVC should be the last option in MHD patients. At present, studies showed that AVF is the main type of vascular access for MHD patients, but the second vascular access type is CVC, and AVG accounts for the lowest proportion in most hospitals of China [27]. In the current study, the proportions of AVF and catheters and artificial blood vessels were, respectively, 70.6% and 29.4%. Autologous arteriovenous fistula (AVF) is considered as the optimal vascular access for most long-term hemodialysis patients, showing better long-term prognosis and lower rates of thrombosis, infectious complications, hospitalization, and mortality compared to other forms of vascular access [28,29,30]. Our research also found that the patients with catheterization and artificial blood vessels as their dialysis path showed a high incidence of anxiety and depression, which might be due to the high risk of chance of infection through the artificial blood vessels, and catheters are often exposed and affect the appearance, which brings about the patient’s worry.

Zhang et al. [31] studied the factors associated with depression and anxiety among 187 patients with ESRD who were undergoing MHD. The results showed that older age was the factor highly related to depression and anxiety. In general, the elderly patients are physically weak, coupled with various underlying diseases and poor immune capacity, which are the factors that contribute to the mental problems in the MHD patients, while there was also limited numbers of individuals accompanying persons for visiting the hospital during the epidemic, which further increased the mental burden of the patients. However, our result showed that during the period of regular prevention and control of COVID-19, age ≥ 50 years, in contrast with those <50 years, was the protective factor for having anxiety and depression in the MHD patients, which might be explained by the fact that many young patients have to bear the heavy responsibility of supporting their families and parents in China. At the same time, because of the epidemic, many working age patients may be at risk of losing their jobs or reducing their remuneration, which increases the mental burden of patients.

Moreover, patients’ monthly income was negatively correlated with the incidence of anxiety. Kidney transplantation is far from sufficient for the treatment of ESRD patients due to the scarcity of renal donors, and hemodialysis is one of the main treatments during the waiting period of kidney transplantation. The patients with CKD need to receive regular dialysis every week to maintain relatively normal metabolism, but the medical costs of hemodialysis are huge. Although hemodialysis could improve symptoms, long-term MHD patients are prone to cardiovascular and cerebrovascular diseases, malnutrition, renal bone disease and other complications. High costs and low income (mainly due to absence from duty) create a double economic burden for the MHD patients, which seriously affects the quality of life and mental health of the patients. Good economic status not only helps patients have good health insurance and social support but also ensures them being able to afford other expenses, such as medicine, transportation, nutritive foods, and emergency expenses. Therefore, patients with lower income might have more concerns and fears about the treatment and prognosis of the disease.

The study showed that the longer duration of dialysis (>3 years) was associated with depression in the MHD patients. Sumida [32] conducted a nationwide registry-based retrospective cohort study of 216,246 patients receiving MHD. The results showed that dialysis patients with longer dialysis duration often have more comorbidities. Along with the decrease in renal function in patients with CKD, a variety of complications may occur, including anemia, hypertension, myocardial ischemia, heart failure, cognitive dysfunction, etc., which could interfere with multiple organs and systems. The metabolic disorders often result in discomfort, such as itchy skin and joint pain, in the MHD patients, which increased their mental burden.

The 2017 KDIGO guidelines suggested that the target level of Hb should be between 110 and 120 g/L. In the current study, only 78.2% of the patients reached this target. Some studies [23,33] have found that Hb was closely related to patient survival. For its increase of each 10 g/L, the mortality of patients was reduced by 10%, while the hospitalization rate was reduced by 12%. In addition, the results of this study suggest that Hb was negatively associated with the incidence of anxiety and depression in the MHD patients. Low levels of Hb may cause dizziness, weakness, palpitation and malnutrition in the patients. The range of normal levels of Hb can improve the patient’s exercise ability and meet the needs of their daily activities. Therefore, it is important for MHD patients to keep the Hb levels within the target range. Controlling Hb in a reasonable range can help patients maintain a good physical and nutritional status to engage in physical exercise and social activities so as to maintain physical and mental pleasure and fight against negative emotions.

Studies have shown that the prevalence of hyperphosphatemia in MHD patients in China is as high as 76%, but the compliance rate is only 38.5% [34].The patients with uremia are often accompanied with hyperphosphatemia, and if it is not well managed, it would cause a metabolic disorder of calcium and phosphorus, which in turn might induce osteoporosis, renal bone disease, pathological fractures and other complications, and more seriously, it could also lead to vascular calcification and cardiac attack. There is an independent correlation between serum phosphorus levels, renal failure and mortality [35]. High serum phosphorus levels could increase the risk of all-cause mortality (20%), renal failure (36%), and renal transplant failure (36%) [36]. The reports indicated that cardiovascular death among patients increase by 50% for every 0.3 mmol/L increase in serum phosphorus levels in patients with CKD [37]. The present study also showed that serum phosphorus was associated with the incidence of depression in the MHD patients. This might be explained as the high level of phosphorus induced itching and joint pain, which caused discomfort and insomnia in the patients.

Multivariate logistic regression analysis also found that PTH was positively associated with the incidence of anxiety and depression in the MHD patients. PTH acts on multiple organs and systems of the human body, high serum PTH levels not only cause skeletal system lesions but also increase cardiovascular events, affect immune system function and impact the nutritional status in MHD patients [38]. The result of the positive correlation between the high level of PTH and incidence of anxiety and depression might be attributed to the toxicity of PTH acting on the central and peripheral nervous systems [39] as well as its influence on calcium and phosphorus metabolic disorders, insomnia and other factors. Insomnia could further aggravate the patient’s anxiety and depression. Well-controlled serum levels of phosphorus and PTH through comprehensive management might delay the progression of CKD, prevent the occurrence and progression of vascular calcification and cardiovascular events, and reduce the occurrence of anxiety and depression in patients with CKD.

## 5. Conclusions

The incidence of anxiety and depression in MHD patients is high even during the period of regular prevention and control of COVID-19 in China. Age, monthly income, feeling of pain, worried about being infected by COVID-19, vascular access of dialysis, blood phosphorus, PTH, and Hb are the factors related to developing anxiety and depression in the MHD patients. The related sectors should pay attention to the mental health status and influencing factors of the patients as well as provide appropriate psychological counseling and intervention to them. Good nutritional support and adjusting the disorder of metabolism, especially the blood concentrations of calcium and phosphorus, is crucial for the MHD patients. The patients with low income and longer duration of dialysis should be paid even great attention in preventing anxiety and depression to improve their prognosis and hopefully enhance their quality of life. Finally, the appropriate assessment and adequate psychological support should be provided to the dialysis patients, especially the MHD patients.

### Limitations of the Study

There are several limitations in this research. First, this was a cross-sectional study, which could not provide a direct cause-and-effect risk association. Second, a cross-sectional study had no follow up information; therefore, there are no longitudinal data available regarding the long-term development of depression/anxiety during the period of regular prevention and control of COVID-19. Finally, the sample size was not large enough, and the subjects were selected from a single dialysis center, which might have created bias regarding the sampling. 

## Figures and Tables

**Figure 1 healthcare-11-00941-f001:**
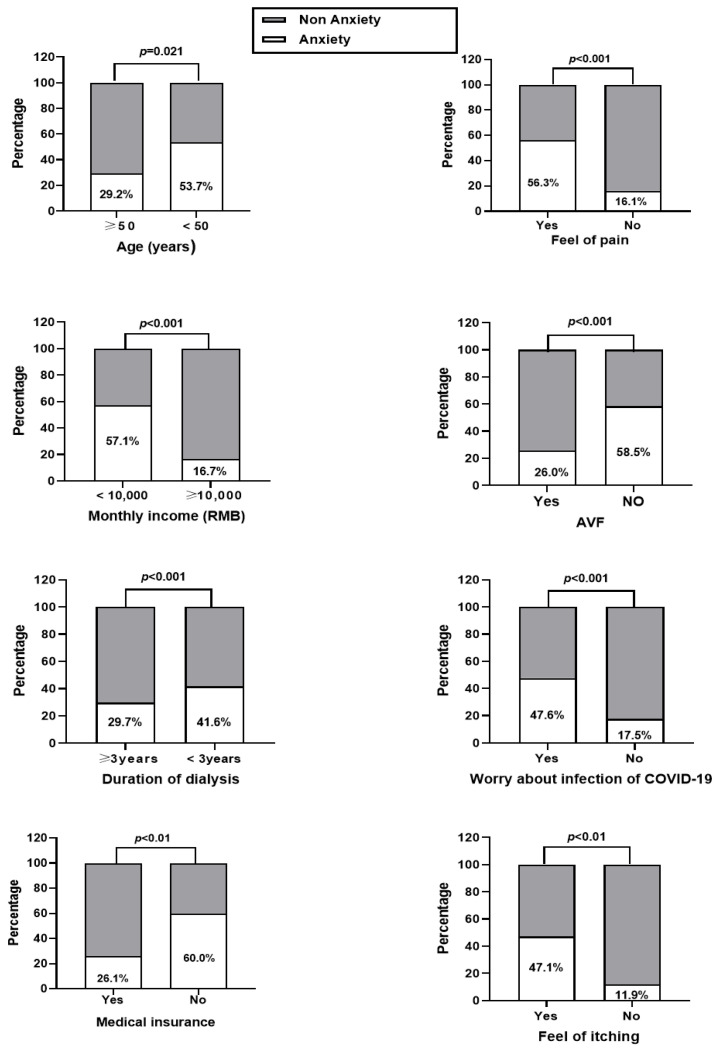
Single-factor analysis of anxiety in the MHD patients with different demographic characteristics.

**Figure 2 healthcare-11-00941-f002:**
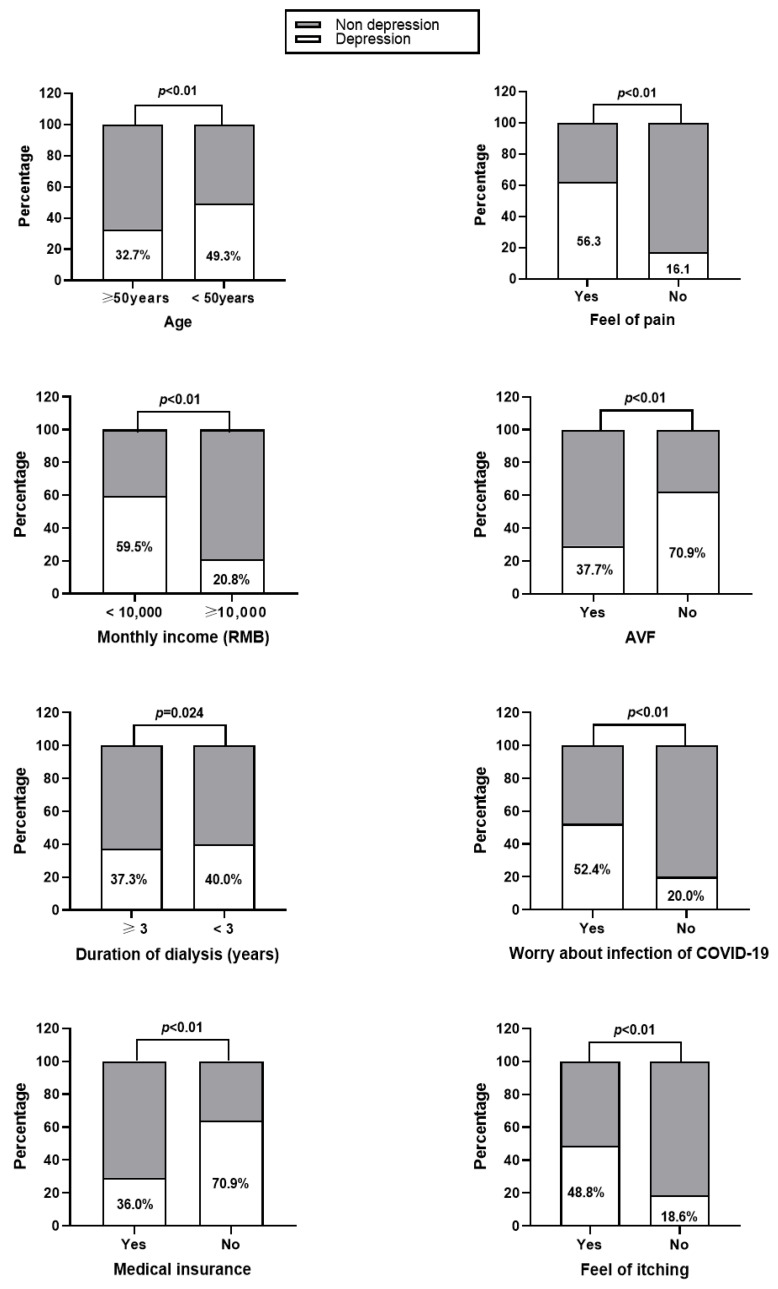
Single-factor analysis of depression in the MHD patients with different demographic characteristics.

**Figure 3 healthcare-11-00941-f003:**
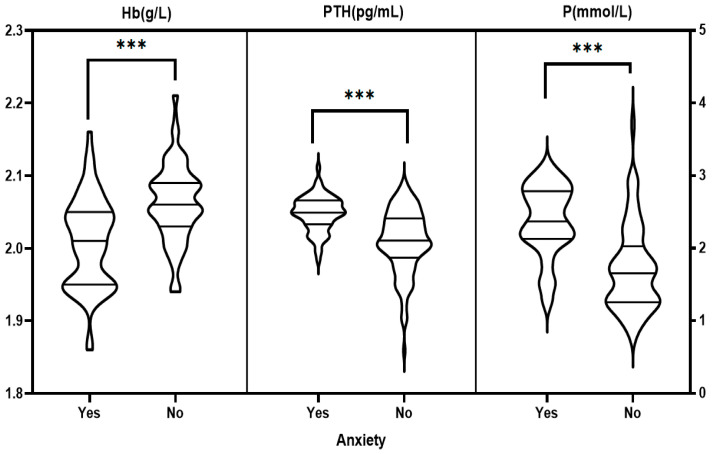
Single-factor analysis of anxiety in the MHD patients with laboratory tests. Note: PTH, Hb values were log-converted. *** *p* < 0.001.

**Figure 4 healthcare-11-00941-f004:**
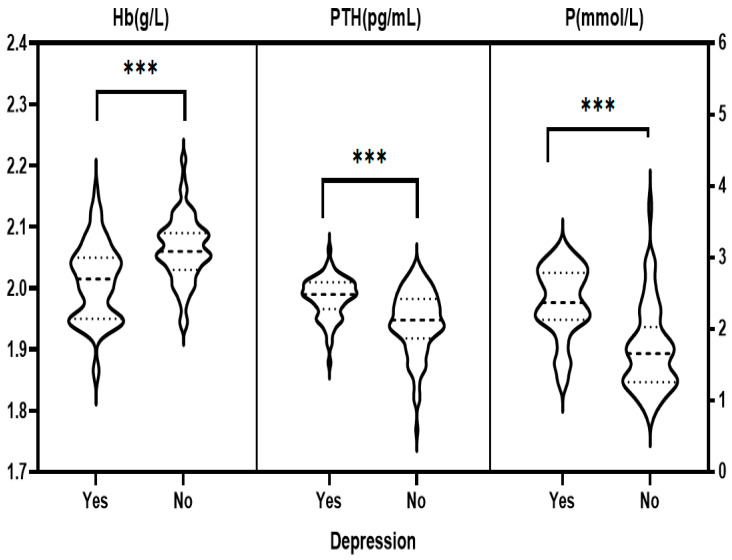
Single-factor analysis of depression in the MHD patients with laboratory tests. Note: PTH, Hb values were log-converted. *** *p* < 0.001.

**Figure 5 healthcare-11-00941-f005:**
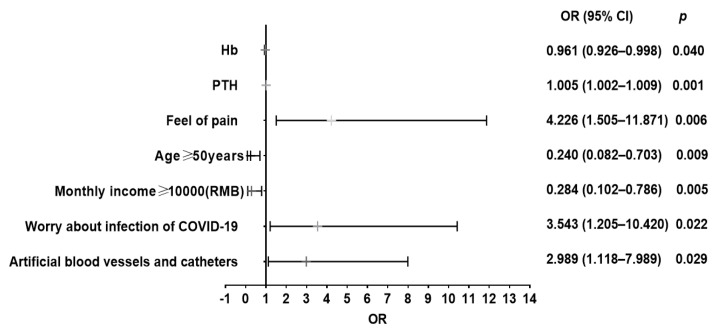
Logistic analysis of risk factors of anxiety in the MHD patients.

**Figure 6 healthcare-11-00941-f006:**
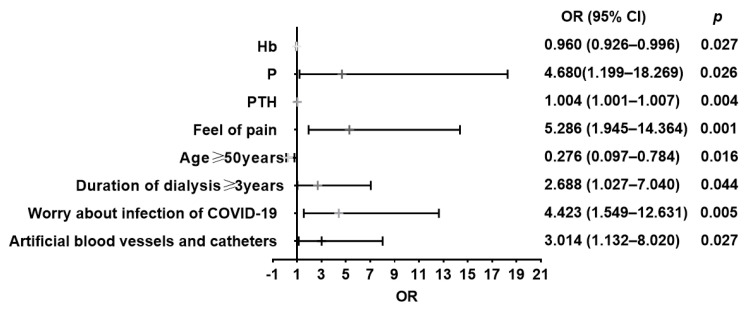
Logistic analysis of risk factors of depression in the MHD patients.

**Table 1 healthcare-11-00941-t001:** Pearson correlation analysis between SAS/SDS score and laboratory tests in the MHD patients (N = 180).

Variables	Mean ± SD	*r* (SAS)	*p* (SAS)	*r* (SDS)	*p* (SDS)
PTH	243.73 ± 190.62	0.395	<0.01	0.353	<0.01
P	1.98 ± 0.64	0.548	<0.01	0.537	<0.01
Hb	111.49 ± 15.58	−0.453	<0.01	−0.465	<0.01

Notes: SDS: Self-Rating Depression Scale; SAS: Self-Rating Anxiety Scale; P: phosphorus (mmol/L); PTH: parathyroid hormone (pg/mL); Hb: hemoglobin (g/L).

## Data Availability

The data obtained in this study are unavailable due to privacy and ethical restrictions.

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
