# Peer review of "The Situation and Influencing Factors of Depression and Anxiety in Patients of Hemodialysis during the COVID-19 Pandemic in China"

_healthcare, 2023, doi:10.3390/healthcare11070941_

Round 1

Reviewer 1 Report

1. The purpose in the abstract is not very much like the purpose, can describe the practical significance of the problem.

2. The statistical methods are too general, please specify the data form and the specific statistical methods used.

3. The results of the text are clearly expressed and the pictures are beautiful; But do the double axes of Figure 3 and Figure 4 indicate the units of the ordinate better? And is the axis unit too large?

4. Why age was grouped according to the criterion of 50 years, and whether the pain was pain during treatment or other conditions.

5. The standard deviation of PTH is half of the mean. Should we use the median to represent the central tendency and change the statistical method? Or do things, like log transformation to make the data normally distributed and then do an analysis of variance?

6. The assignment of logistic regression and the results of regression analysis model can be presented in the form of attached tables.

7. Less elaboration of the results of P and Hb was given in the discussion.

Author Response

Dear Reviewer:

Thank you for your valuable comments. We have carefully gone through the comments and made appropriate correction. Hopefully, the revised version would be improved significantly.  Please find the following point to point response to the reviewer’s comments: 

  1. The purpose in the abstract is not very much like the purpose, can describe the practical significance of the problem.

Thanks for your questions and suggestions. We modified as the following.

To investigate the incidence of depression and anxiety among maintenance hemodialysis (MHD) patients during the regular prevention and control stage of COVID-19 in China, and the influencing factors. (Please see the manuscript, page 1, line 9-11).

  1. The statistical methods are too general, please specify the data form and the specific statistical methods used.

Thanks for your question and suggestion. We modified as the following.

All data were analyzed by using SPSS 22.0 (SPSS Inc., Chicago, IL, USA). Continuous variables were presented as mean ± SD. Categorical variables were expressed as number and percentages (%). Differences between groups were compared using one-way ANOVA and Chi-square test. Spearman's correlation coefficient (r) was used to assess the association between laboratory test data and levels of anxiety and depression. Statistically significant variables were further analyzed through multivariate logistic regression. Estimates of association strength were demonstrated by odds ratios (ORs) with 95% confidence intervals (CIs). Significant levels were set as P<0.05 (two-tailed). (Please see page 3, line 110-118).

  1. The results of the text are clearly expressed and the pictures are beautiful; But do the double axes of Figure 3 and Figure 4 indicate the units of the ordinate better? And is the axis unit too large?

Thanks for your questions and suggestions.

The PTH value and Hb values were log transformed, and were normal distributed. The values of the blood PTH and P were close and suitable to share one axis (on the right) , and Hb level was differed from the two, therefore, the second axis (on the left) was employed. (Please see the revised figure 3 and 4 in section 3.3 page 7).

  1. Why age was grouped according to the criterion of 50 years, and whether the pain was pain during treatment or other conditions.

Thanks for your questions and suggestions.

The study subjects were middle and up age individuals, according to the age distribution of the sample, 50 years old was selected as the cut-off point. The anxiety and depression of the patients were scored according to the patient's feeling during the survey week, so the feeling of pain (regarding the dialysis procedure and/or disease status) in the patient was also recorded at the time of survey. (Please refer page 3, line122 and line 124)

  1. The standard deviation of PTH is half of the mean. Should we use the median to represent the central tendency and change the statistical method? Or do things, like log transformation to make the data normally distributed and then do an analysis of variance?

Thanks for your questions and suggestions.

    The data of PTH and Hb were processed via Log transformation and the figures were revised. (Please refer Figure 3 and Figure 4 in section 3.3 page7)

  1. The assignment of logistic regression and the results of regression analysis model can be presented in the form of attached tables.

Thanks for your questions and suggestions. We provided the Appendix to show the detailed data. (Please refer page 15).

Appendix 1.  The assignment of logistic regression

Item

Assignment

Duration of dialysis≥ 3years

1:Duration of dialysis≥ 3years; 0:Duration of dialysis< 3years

Age≥ 50years

1:Age≥ 50years; 0:Age< 50years

PTH(pg/mL)

Continuity variables

Hb(g/L)

Continuity variables

Feel of pain

1:Yes;   0:No

Worry about infection of COVID-19

1:Yes;   0:No

Artificial blood vessels and catheters

1:Artificial blood vessels and catheters;    0:AVF

P(mmol/L)

1:Normal group;  0:Abnormal group

Monthly income≥10000(RMB)

1:Yes;   0:No

Appendix 2.  Logistic analysis of risk factors of anxiety in the MHD patients

β

SE

Wald

P

OR

95% CI

Age≥ 50years

-1.427

0.548

6.774

0.009

0.24

0.082-0.703

PTH(pg/mL)

0.005

0.002

10.417

0.001

1.005

1.002-1.009

Hb(g/L)

-0.039

0.019

4.228

0.04

0.961

0.926-0.998

Feel of pain

1.441

0.527

7.483

0.006

4.226

1.505-11.871

Monthly income≥10000(RMB)

-1.261

0.52

5.873

0.015

0.284

0.102-0.786

Worry about infection of COVID-19

1.265

0.55

5.283

0.022

3.543

1.205-10.42

Artificial blood vessels and catheters

1.095

0.502

4.762

0.029

2.989

1.118-7.989

Appendix 3.  Logistic analysis of risk factors of depression in the MHD patients

β

SE

Wald

P

OR

95% CI

Duration of dialysis≥ 3years

0.989

0.491

4.054

0.044

2.688

1.027-7.04

Age≥ 50years

-1.286

0.532

5.839

0.016

0.276

0.097-0.784

PTH(pg/mL)

0.004

0.002

8.177

0.004

1.004

1.001-1.007

Hb(g/L)

-0.04

0.018

4.86

0.027

0.96

0.926-0.996

Feel of pain

1.665

0.51

10.659

0.001

5.286

1.945-14.364

Worry about infection of COVID-19

1.487

0.535

7.711

0.005

4.423

1.549-12.631

Artificial blood vessels and catheters

1.103

0.499

4.879

0.027

3.014

1.132-8.02

P(mmol/L)

1.543

0.695

4.934

0.026

4.68

1.199-18.269

  1. Less elaboration of the results of P and Hb was given in the discussion.

Thanks for your suggestions. We did modification as following.

The 2017 KDIGO guidelines suggested that the target level of Hb should be between 110 and 120 g/L. In current study, only 78.2% of the patients reached this target. Some studies [23,33] have found that Hb was closely related to patient survival. For its increase of each 10 g/L, the mortality of patients was reduced by 10%, while the hospitalization rate was reduced by 12%. In addition, the results of this study suggest that Hb was negatively associated with the incidence of anxiety and depression in the MHD patients. Low level of Hb may cause dizziness, weakness, palpitation and malnutrition in the patients. Range of normal levels of Hb can improve the patient’s exercise ability and meet the needs of the their daily activities. Therefore, it is important for MHD patients to keep the Hb levels within the target range. Controlling Hb in a reasonable range can help patients maintain a good physical and nutritional status to engage in physical exercise and social activities, so as to maintain physical and mental pleasure and fight against negative emotions.

Studies have shown that the prevalence of hyperphosphatemia in MHD patients in China is as high as 76%, but the compliance rate is only 38.5% [34].The patients with uremia are often accompanied with hyperphosphatemia, and if it is not well managed, it would cause metabolic disorder of calcium and phosphorus, which in turn, might induce osteoporosis, renal bone disease, pathological fractures and other complications, and more serious, it could also lead to vascular calcification and cardiac attack. There is an independent correlation between serum phosphorus levels, renal failure and mortality [35]. High serum phosphorus levels  could increase the risk of all-cause mortality (20%), renal failure (36%), and renal transplant failure (36%) [36]. The reports indicated that cardiovascular death among patients increase by 50% , for every 0.3 mmol/L increase in serum phosphorus levels in patients with CKD [37]. The present study also showed that serum phosphorus was associated with the incidence of depression in the MHD patients. This might be explained as the high level of phosphorus induced itching and joint pain, which caused discomfort and insomnia in the patietns.

Multivariate logistic regression analysis also found that PTH was positively associated with the incidence of anxiety and depression in the MHD patients. PTH acts on multiple organs and systems of the human body, high serum PTH level not only causes skeletal system lesions, but also increases cardiovascular events, affects immune system function and nutritional status in MHD patients [38]. The result of positive correlation between high level of PTH and incidence of anxiety and depression might attribute to the toxicity of PTH, acting on the central and peripheral nervous systems [39], as well as its influence on calcium and phosphorus metabolic disorders, insomnia and other factors. Insomnia could further aggravate the patient's anxiety and depression. Well controlled serum levels of phosphorus and PTH through comprehensive management might delay the progression of CKD and prevent the occurrence and progression of vascular calcification and cardiovascular events and reduce the occurrence of anxiety and depression in patients with CKD.

 (Please see in section 4.2 page 11-12, line 315-355).

Reviewer 2 Report

The study is interesting, manuscript is well written.

Some comments:

1) Please present percentaged with one digit after comma, for example 29.2% rather than 29.204%.

2) In methods you write " SAS is reliable, efficient, and widely used. Cronbach's α coefficient is 0.823".  What does it mean? And why is it important?

3) Interretation ; there are no causal realtionships. You should use terms like associations . For example "The study suggested that the longer duration of dialysis (> 3 years) was the independent factor for depression" is wrong.  Correct is: longer duration of dyalisis was asscocited withdepression".  This should be checked n full mansucript.

4) Further study limitation is that data were taken from one center (hopsital) only, as well as that country from defined country cannot be extrapolate to other countries.

Author Response

Dear Reviewer:

Thank you for your valuable comments. We have carefully gone through the comments and made appropriate correction. Hopefully, the revised version would be improved significantly. Please find the following point to point response to the reviewer’s comments:

1) Please present percentaged with one digit after comma, for example 29.2% rather than 29.204%.

Thanks for your questions and comments. In the manuscript, we kept one digit after point. Please see 3.1 page 3, line 121-131).

  • In methods you write " SAS is reliable, efficient, and widely used. Cronbach's α coefficient is 0.823". What does it mean? And why is it important?

Thanks for your questions and comments.

The Cronbach's α coefficient is an indicator that evaluates the reliability of the internal consistency of multi-item rating scale. Scoring scales are widely used in psychology and the social sciences to address so-called latent structures that could not be measured directly, such as self-esteem, anxiety, depression, somatization, etc. Cronbach's α reflect the degree of correlation or interrelationship between the scale entries, and its value is usually between 0 and 1, with values closer to 1 indicating stronger internal consistency. Cronbach's α coefficient > 0.7, which could be considered having good agreement between entries.

3) Interrelation; there are no causal relationships. You should use terms like associations. For example "The study suggested that the longer duration of dialysis (> 3 years) was the independent factor for depression" is wrong. Correct is: longer duration of dialysis was associated with depression". This should be checked in full manuscript.

Thanks for your questions and comments. We did modification. Please see 4.2 page 11, line 306-307 and page 12, line 339 and line343-344.

  • Further study limitation is that data were taken from one center (hospital) only, as well as that country from defined country cannot be extrapolate to other countries.

Thanks for your questions and comments. Yes, it is the limitation of the study. Please refer page 13 , line 377-378.

Round 2

Reviewer 1 Report

Agree to modify the revised version and agree to publish the manuscript.

Reviewer 2 Report

N/A